# Good Vibrations: Structural Remodeling of Maturing Yeast Pre-40S Ribosomal Particles Followed by Cryo-Electron Microscopy

**DOI:** 10.3390/molecules25051125

**Published:** 2020-03-03

**Authors:** Ramtin Shayan, Dana Rinaldi, Natacha Larburu, Laura Plassart, Stéphanie Balor, David Bouyssié, Simon Lebaron, Julien Marcoux, Pierre-Emmanuel Gleizes, Célia Plisson-Chastang

**Affiliations:** 1Laboratoire de Biologie Moléculaire Eucaryote, Centre de Biologie Intégrative, Université de Toulouse, CNRS, UPS, 118 route de Narbonne, 31062 Toulouse CEDEX, France; r.shayan@mail.cryst.bbk.ac.uk (R.S.); dana.rinaldi@ibcg.biotoul.fr (D.R.); n.larburu@imperial.ac.uk (N.L.); Laura.plassart@ibcg.biotoul.fr (L.P.); stephanie.balor@ibcg.biotoul.fr (S.B.); simon.lebaron@inserm.fr (S.L.); 2Institut de Pharmacologie et Biologie Structurale, Université de Toulouse, CNRS, UPS, 205 route de Narbonne, 31062 Toulouse CEDEX, France; david.bouyssie@ipbs.fr

**Keywords:** ribosome assembly, small ribosomal subunit, bottom-up proteomics, cryo-EM, single particle analysis

## Abstract

Assembly of eukaryotic ribosomal subunits is a very complex and sequential process that starts in the nucleolus and finishes in the cytoplasm with the formation of functional ribosomes. Over the past few years, characterization of the many molecular events underlying eukaryotic ribosome biogenesis has been drastically improved by the “resolution revolution” of cryo-electron microscopy (cryo-EM). However, if very early maturation events have been well characterized for both yeast ribosomal subunits, little is known regarding the final maturation steps occurring to the small (40S) ribosomal subunit. To try to bridge this gap, we have used proteomics together with cryo-EM and single particle analysis to characterize yeast pre-40S particles containing the ribosome biogenesis factor Tsr1. Our analyses lead us to refine the timing of the early pre-40S particle maturation steps. Furthermore, we suggest that after an early and structurally stable stage, the beak and platform domains of pre-40S particles enter a “vibrating” or “wriggling” stage, that might be involved in the final maturation of 18S rRNA as well as the fitting of late ribosomal proteins into their mature position.

## 1. Introduction

In every living cells, ribosomes are RNP particles that translate genomic information into proteins. These huge molecular machines are composed of a small and a large subunit, carrying the decoding center and peptidyl transferase activity, respectively. In eukaryotes, small (40S) ribosomal subunits are composed of 1 ribosomal RNA (18S rRNA) and 33 ribosomal proteins (RPS), while large ribosomal subunits are composed of 3 ribosomal RNAs (25/28S, 5.8S and 5S rRNAs) and 46 ribosomal proteins (RPL). Forming functional ribosomal subunits is an extremely complex and energy demanding process, which occurs sequentially and requires the transient association of 200 ribosome biogenesis factors (RBFs). These RBFs are proteins that drive pre-rRNA cleavage, modification and folding, as well as RPs chaperoning and integration into nascent ribosomal subunit precursors. Although they were quasi-exhaustively identified two decades ago by genetic screens or proteomics studies [1,2,3], and have been predicted to harbor various activities such as NTPase, nuclease, helicase or kinase based on their secondary structures (for reviews, [4,5,6,7]), the precise role of most RBFs in ribosome biogenesis remains elusive.

In yeast, assembly of the small ribosomal subunit starts in the nucleolus, within large complexes called 90*S* particles or SSU processome [8]. Endonucleolytic cleavages of pre-rRNA release so-called pre-40S particles, which are composed of almost all RPS [9] and the 20S pre-rRNA, a precursor to the 18S rRNA bearing a 200-long nucleotide extension at its 3′ end. This pre-40S particle, to which only a few RBFs are associated, is rapidly exported to the cytoplasm [10].

Once there, RBFs escorting cytoplasmic pre-40S sequentially dissociate from maturing particles. Though the precise timing of release of these RBFs is still largely unclear, cytoplasmic maturation of yeast pre-40S particles can be subdivided into three successive steps:

(1) Pre-40S particles just exported from the nucleus are called early cytoplasmic particles. They are composed of the 20S pre-rRNA, all RPS with the exception of Rps31, Rps10 and Rps26, and the RBFs Enp1, Ltv1, Tsr1, Rio2, Dim1, Dim2/Pno1 and Nob1 [3,11]. UV Cross-linking and analysis of cDNA (CRAC analyses) [12,13] as well as cryo-EM studies [14,15,16,17], have allowed to precisely position most RBFs on these early cytoplasmic pre-40S particles. Their 3D structure resembles that of mature small ribosomal subunits, but with an “open” conformation of pre-rRNA that is thought to prevent premature entry of pre-40S particles into the translation cycle. Furthermore, most of the identified RBFs physically block the association of the translation machinery by occupying functional sites on the 40S subunit. Dim1, which is responsible for base methylation of adenosines 1781 and 1782 on the 18S rRNA [18], was located on the intersubunit side under the platform region by low resolution cryo-EM studies, but seems to be absent or structurally instable in most of the high resolution 3D structures of yeast pre-40S particles published to date [16,17,19], suggesting an early dissociation from cytoplasmic pre-40S particles. Nob1, the endonuclease responsible for the final cleavage of the 20S pre-rRNA into mature 18S rRNA [20], could not be precisely located on any EM structures of yeast pre-40S particles. Only in human pre-40S particles (either TAP-tag purified or in vitro reconstituted) could NOB1 be positioned on the platform region. In these particles, NOB1 endonucleolytic activity is thought to be impeded because its active site is too far away from its substrate (i.e., 18S rRNA 3′ end), which is further obstructed by the presence of DIM2/PNO1 [21,22]. Within early cytoplasmic pre-40S particles, Ltv1 and Enp1 (located on the beak of the particle) are phosphorylated by Hrr25, the yeast ortholog to human Casein Kinase 1 (CK1). This leads to the dissociation of Ltv1, which in turn could provoke the release of Enp1 [11,14,15,23]. Dissociation of Ltv1/Enp1 would allow their replacement by Rps10 at the same position on the beak of the pre-40S particle. Moreover, this would also permit Rps3, located between the head and the beak of the pre-40S, to be fitted into its mature position [24,25]. This structural modeling of the beak region would allow the pursuit of the small subunit maturation [19].

(2) Release of Ltv1 and Enp1 gives rise to pre-40S particles termed intermediate cytoplasmic particles, to which Tsr1, Rio2, Nob1 and Dim2 would remain stably bound. Recycling of Tsr1 and Rio2 (both positioned on the intersubunit side and obstructing the mRNA groove) would require the activity of serine kinases Rio2 and/or Rio1, although their precise role and time of action remain to be determined [26,27].

(3) Dissociation of Tsr1 and Rio2 would yield late pre-40S particles, in which only Dim2 and Nob1 would be stably bound. In these, a remodeling of the platform region driven by Rio1 activity would occur, which would allow the incorporation of Rps26 in its mature position, and Nob1 to cleave the 20S pre-rRNA at site D, giving rise to mature 18S rRNA [13,27,28]. These last steps might occur in so-called 80S-like particles, in which pre-40S particles would assemble to 60S subunits and mimic translation initiation cycle for a quality control checkpoint of the small ribosomal subunit maturation [13,29,30].

Interestingly, a study based on high-throughput rRNA structure probing (ChemModSeq) revealed that pre-40S particles containing Rio2, Tsr1, Ltv1 and Enp1 harbored a 3′ major domain (corresponding to the beak and platform domains of the small ribosomal subunit) with a higher degree of flexibility than late pre-40S particles, which lacked these proteins [28]. Furthermore, a very recent cryo-EM study performed on pre-40S particles with a truncated form of Rps20 revealed that these pre-40S particles could be trapped in at least two structural classes, one with immature beak and platform domains, and the second with a fully matured head and a highly dynamic platform [19]. Taken together, this suggests that pre-40S particles might undergo various states ranging from structurally stable to highly dynamic, and that the platform might be the last domain to be matured.

Despite this wealth of information given by biochemical and functional studies, as well as structural ones for early cytoplasmic pre-40S particles, little is known regarding the fate of cytoplasmic pre-40S particles after the earliest step, and the timing of events presiding over their final maturation. In order to better characterize the structural remodeling steps leading from early to intermediate cytoplasmic pre-40S particles, we have performed proteomics and cryo-EM studies of pre-40S particles purified from *Saccharomyces cerevisiae* using a tagged version of Tsr1 as bait, in a wild-type phenotype context. Our results indicate that particles can be subdivided into several structural classes, all of them harboring an almost mature body. The first one, overall structurally stable, likely correspond to an early maturation state. Other pre-40S particles, which we attribute to intermediate maturation states, would see first their beak and then their platform enter a highly dynamic (“vibrating” or “wriggling”) state. Based on these observations and previously published results, we hypothesize that structural instability of these domains might reflect the remodeling events associated with correct maturation of 20S pre-rRNA and final RPS assembly. These structural results reinforce the idea that RBFs, besides their putative enzymatic activities, help stabilizing pre-rRNA into an immature conformation, thus preventing incompetent pre-40S particles to enter into the translating pool of small ribosomal subunits.

## 2. Results

### 2.1. Proteomic Characterization of Tsr1-purified pre-40S Particles

All previous structural investigations on late/cytoplasmic precursors to the small ribosomal subunit, whether prokaryotic or eukaryotic ones, hinted at their wide structural and compositional heterogeneity [11,14,15,16,17,21,22,31,32]. In order to reduce this heterogeneity, we prepared cytoplasmic pre-40S particles by a 3-steps purification scheme. Using the cytoplasmic RBF Tsr1 as bait (Tsr1-FPZ) [33], we performed a tandem affinity purification, followed by a physical separation step on a 10%–30% sucrose gradient. The sedimentation profile showed that the majority of Tsr1-FPZ pre-40S particles presented a peak position very similar to that of mature 40S ribosomal subunits, thus migrating as “free” pre-40S subunits. A smaller peak of heavier fractions indicates that Tsr1-FPZ could be found in bigger complexes, some of them probably being 80S-like particles [29,30,34,35] (Figure 1a, upper panel). Transmission electron microscopy (TEM) observations after negative staining confirmed that the particles sedimenting on lighter fractions had the overall size and shape of 40S subunits, while heavier fractions contained larger, rounder objects, with dimensions compatible with 80S-like particles (Figure 1a, lower panel).

Pre-40S particles collected from lighter gradient fractions (hereafter called “Tsr1-FPZ pre-40S particles”) were analyzed for their protein composition by SDS-PAGE (Figure 1b), and bottom-up proteomics (Figure 1c). Label-free quantification of the relative abundances of the identified proteins indicated that these pre-40S were composed of all RPS with the exception of Rps26 and Rps31. Rps10 was identified by only one tryptic peptide, suggesting that association of this protein to pre-ribosomal subunits is still highly dynamic at this maturation step (Figure 1c, proteins with a ratio of observed/observable peptides <30% are marked in red). Ribosome biogenesis factors Tsr1, Rio2, Enp1, Ltv1, Dim2 and Nob1 were found associated to pre-40S particles; however, only traces of Dim1 were found in the pool of Tsr1-FPZ pre-40S particles. We did not exclude that our purification protocol might artificially dissociate Dim1 from pre-40S particles, but other studies have suggested that this RBF has already left pre-ribosomes at this maturation stage (see below).

Bottom-up proteomics also allowed us to detect phosphorylation sites on several RBFs, namely Tsr1, Ltv1 and Enp1 (Table 1). Tsr1 was found to be phosphorylated on residues T354 or S358, both located on its domain II [36]. The corresponding non-phosphorylated peptide could not be found, suggesting that 100% of Tsr1 would harbor such a modification; the kinase responsible for this phosphorylation remains to be identified. Hrr25 is the kinase that phosphorylates Ltv1 in both human and yeast cells [11,19,37], and systematic point mutations identified Ltv1 Serines S336, S339 and S342 as its putative phosphorylation targets [23]. Two of these sites were retrieved in our bottom-up analyses, which revealed phosphorylation marks on S336 or S339, though for a small fraction (6.5%) of the considered peptides (Table 1).

Like for Ltv1, Enp1 phosphorylation has also been shown to be mediated by CK1/Hrr25 in human and yeast cells [11,37], although its phosphorylated residues have not been precisely mapped to date. Our bottom-up analyses revealed phosphorylation marks on S172, however, like for Ltv1, the amount of phosphorylated peptide seems to be low (2.1% abundance compared to the non-phosphorylated corresponding peptide); other amino acids of Enp1 also bore phosphorylation marks, but in negligible amounts (Table 1). As these modifications are thought to trigger Ltv1 and Enp1 release from cytoplasmic pre-40S particles [12,23,37], the small amount of phosphorylated Ltv1 and Enp1 associated to Tsr1-FPZ pre-40S particles suggests that their phosphorylation might be rapidly followed by the release of these RBFs from pre-ribosomal particles, as recently suggested [19].

Finally, bottom-up proteomic analyses did not reveal the presence of Rio1, which is involved in late cytoplasmic maturation steps of the small ribosomal subunit. Taken together, these results confirm that the purification bait and scheme that we used allowed us to purify cytoplasmic pre-40S particles at early and intermediate cytoplasmic maturation stages.

### 2.2. Cryo-EM Analysis of Tsr1-FPZ pre-40S Particles

To better describe the remodeling events occurring to pre-40S particles in the cytoplasm, we then performed a single particle cryo-EM analysis of Tsr1-FPZ pre-40S particles. A first round of image analysis allowed us to obtain a “consensus” 3D structure, with an overall resolution of 3.1 Å according to RELION’s gold standard Fourier Shell Correlation (FSC) (see Material and methods). Visual Inspection of the 3D volume as well as local resolution assays confirmed that the body of the particle was very well resolved (Appendix A), and thus presented little flexibility or variability. Interestingly, in this region of the EM density map, previously described ribose 2-O’ methylations (for review, [38]) were clearly visible for adenosines A100, A420, A436 and A796 (Figure 2). Other rRNA modifications are located in less well resolved regions of the map, and were thus not added to the final atomic model. For instance, like for other existing pre-40S structures, the region located between the upper part of rRNA h44 and the platform was not well resolved in our cryo-EM maps. This prevented us from checking whether dimethylation of adenosines A1780 and A1781 performed by Dim1 [18] was present on the Tsr1-FPZ pre-40S particles. However, primer extension analyses performed on early, intermediate and late cytoplasmic pre-40S particles showed that this modification has already occurred at all these stages [28,30]. Based on these previous studies and the absence of Dim1 in our bottom-up proteomics analyses, we speculated that, in wild-type conditions, rRNA dimethylation by Dim1 and its subsequent dissociation from the particles might occur upstream of the maturation stage corresponding to the Tsr1-FPZ pre-40S particles purified here (see discussion).

In stark contrast to the body core, the head harbored much more blurred features, suggesting a high degree of structural heterogeneity. To assess this phenomenon, we first performed global classification assays, which yielded three 3D structures (hereafter termed GC1 to GC3) with overall resolutions ranging between 3.4 and 3.8 Å (Appendix A). In line with the initial consensus 3D structure, comparisons of these classes showed no variations in the body region, and a stable positioning of Tsr1 and Rio2 on the 60S interface side of the particle. The back of the head of the particle also displayed very little variability. Strikingly, only class GC1, corresponding to 26% of the analyzed particles, resembled early cytoplasmic pre-40S structures and displayed densities identified as Enp1 and Ltv1. The two other classes harbored a beak with much more blurred features, which were hardly interpretable. Similarly, classes GC1 and GC2 (representing 26% and 31% of the analyzed particles, respectively) showed a well-defined platform region, with clear features for Dim2, Rps1 and Rps14, while class GC3 displayed a totally blurred platform (Appendix A, compare central sections).

### 2.3. Structural Heterogeneity of the Head Domain

We then reasoned that blurring of the head and platform regions might be at least partly due to their flexibility, such as the well characterized swiveling movement of the head compared to the body [39]. Since RELION performs alignments of the largest domains of the objects of interest (here, the body of the pre-40S particle, which represents 2/3 of its size), this will lead to misalignment, i.e., blurring of more mobile domains (in our case, the head and platform regions) when computing cryo-EM 3D maps. To overcome this issue, we used the particles included in the consensus reconstruction and performed focused 3D classification of the head domain, with signal subtraction of the body and platform regions. This approach yielded three distinct structural classes of the pre-40S head (Appendix A, right panel). The first one hereafter called H1 (Figure 3a), representing 36% of the analyzed particles, was refined up to a resolution of 3.8 Å and appears very similar to previously published early cytoplasmic particles [16,17]. Notably, the rRNA three-way junction formed by helixes h34, h35 and h38 at the base of the head [39] is in an immature, “open” conformation. Amino acids 205–466 of Enp1, as well as and 351–405 of Ltv1 could clearly be positioned on the beak region. The resolved part of Ltv1 forms an elongated structure running throughout the full length of Enp1, extending towards the 60S-interface on the tip of the beak of the pre-40S particle. Fragmented density on the tip of the beak was attributed to Rps12 (eS12), which might not be fully stabilized at this maturation step of the small ribosomal subunit. Similarly, fragmented densities were attributed to amino acids 21–46 of Rps29 and to the globular domain of Rps20 close to their mature position, next to rRNA helices h31 and h41, respectively. Despite its unambiguous detection by bottom-up proteomics, no clear density could be attributed to the presence of Rps3 on this cryo-EM map. This suggests that while they are associated to pre-40S particles in the cytoplasm, structural stabilization is not complete for Rps20 and Rps29, and has not yet occurred for Rps3. Unattributed densities on the tip of the beak occupy the location of Rps31 (violet density on Figure 3a). Since they are positioned in close vicinity of Ltv1 and Enp1, which atomic structures are not fully solved, we hypothesized that the unattributed density might correspond to one of these two RBFs, which might play the role of placeholder for Rps31.

A second 3D class (called H2), encompassing 20% of the particles was refined to an overall resolution of 6.4 Å (Figure 3b). Inspection of the central section of this EM map as well as local resolution assays revealed that the back of the head (located around rRNA helix h39es9) harbored much more distinguishable features than the beak region, suggesting a higher flexibility of the beak compared to the back of the head. Such a variation in subdomains rigidity was less obvious in 3D class H1 (Appendix A). Rigid body docking of the atomic model derived from 3D class H1 into H2 EM map revealed no major structural rearrangement (Figure 3, compare panels a and b). Interestingly, in H2 pseudo atomic model, the C-terminal part of Ltv1 appeared further apart from Enp1, and its last alpha-helix (amino acids 390–405) was rotated from 90° compared to its position in the H1 one (Figure 3c). Our bottom-up proteomic analyses show that only a minor (2%–6%) fraction of Ltv1 and Enp1 are phosphorylated. This seems to indicate that the repositioning of Ltv1 and the wriggling of the beak would not be caused by Ltv1 and Enp1 phosphorylation by Hrr25. Instead, we proposed that this conformational change could be an effect of the interaction between Rps20 and N-terminal residues of Rps3, which precedes Ltv1 phosphorylation, as previously shown by genetic and biochemical approaches [19] (see discussion).

Surprisingly, about 44% of analyzed pre-40S particles were sorted into a third 3D class, with overall blurred features, that could not be auto-refined (3D Class H3 on Appendix A). Of note, local classifications around the head domain, but without signal subtraction, yielded similar results (data not shown). Although we cannot rule out that artifacts due to sample preparation or handling could lead to destabilization or destruction of the head domain, our observations might also indicate that this region is indeed more dynamic than the body. As instability of the beak domain was also previously observed by RNA chemical probing (ChemModSeq) studies [28], we speculated that after a first (early) step where it is structurally stable, the head domain of cytoplasmic pre-40S particles enters a more unstable, “wriggling” stage, that might be a true structural feature of intermediate pre-40S particles.

### 2.4. A “Wriggling” of the Platform Domain

To further investigate structural heterogeneity of the platform, we used the same approach of focused classifications with signal subtraction as described for the head domain (see Appendix A, left panel). This yielded a first 3D class, representing 35% of the particles population, which was auto-refined up to an overall resolution of 3.7 Å. This 3D class, hereafter called P1, unambiguously featured Rps1, Rps14, rRNA helix 23 and Dim2 resting on top of the 3′ end of 18S rRNA (Figure 4a). P1 appeared very similar to other existing yeast early cytoplasmic pre-40S structures [16,17], and Nob1, which is known to directly interact with Dim2 [40,41] could not be detected. Similarly, the ITS1 region of the 20S pre-rRNA made of 200 nucleotides after the 3′ end of the mature rRNA, could not be seen on this structure, suggesting its high flexibility.

A second class, P2, representing 12% of the population, could only be auto-refined to a resolution of 9.1 Å. P2 showed well defined structural features for Rps0 and rRNA helix 26es7, while the platform domain formed by Rps1, Rps14 rRNA h23 was slightly blurred and the 18S rRNA 3′ end was undetectable, indicating structural instability of this region (Figure 4b).

A third class, P3, representing 53% of the particles, could not be refined at all with this approach, probably because images of this class harbored signals not similar enough to be aligned together (see left panel of the central sections in Appendix A). Of note, masked classifications (without signal subtraction) focused around the platform region yielded three classes where the domain formed by Rps1, Rps14, rRNA helix 23 and Dim2 was either fully structured, or partially or totally absent from the platform (data not shown). There again, platform instability might result from purification and/or cryo-EM grid preparation conditions. However, because vibration or unfolding of the platform has already been observed on various precursors to the small ribosomal subunit [19,28] (Rai et al. Biorxiv https://doi.org/10.1101/617910), we made the hypothesis that like for the head domain, after a maturation step where the platform is structurally stable, a signal of an unknown nature could trigger its wriggling. This structural instability of the platform might reflect the final steps of cytoplasmic pre-40S particles maturation (see discussion below).

## 3. Discussion

Cryo-EM is a very powerful method to explore structural heterogeneity within macromolecular complexes, and has widely contributed to understanding ribosome biogenesis in the past few years (for review, [42]). In this work, we studied yeast pre-40S particles purified using Tsr1-FPZ as bait within an otherwise wild-type genetic background. Bottom-up proteomics allowed us to precisely describe the proteic composition of these particles, and our experiments lead us to conclude that, as previously reported in external studies [13,28], they were purified at early and intermediate cytoplasmic maturation stages.

### 3.1. Cryo-EM Structures Suggest that pre-40S Particles Transit through a Vibrating State

Our cryo-EM analysis shows that these pre-40S particles have a structurally stable body resembling that of a mature subunit, with the exception of the upper part of h44 that is detached from the body, as already described for cytoplasmic pre-40S particle structures. Focused classifications with signal subtraction [43] revealed that both the beak and platform domains could be subdivided into several structural classes. For each domain, one class was structurally stable and well resolved (H1 and P1), and closely resembled already published pre-40S structures purified at early cytoplasmic maturation states [15,16]. The other classes (H2 and P2 and H3 and P3) were gradually less well resolved, and sections of the 3D reconstructions revealed an increased blur of these domains (see Appendix A) that we referred to as “wriggling” or “vibrating” states. Accordingly, in global classification assays, Tsr1-FPZ structures ranged from well resolved, overall structurally stable and very similar to already published early pre-40S particles structures [16], to increasingly blurred for the head and the platform domains (see GC1-GC3 in Appendix A).

We could not rule out that this structural instability is an artifact caused by purification and/or cryo-EM grid preparation. However, several facts lead us to propose that transient vibration of the beak and platform might actually take place within cells.

First, several studies performed both in prokaryotic and eukaryotic cells using either *in vitro* reconstitution of ribosomal subunits for the former or systematic deletion of ribosomal proteins for the latter, suggested that the head and platform are the last regions to be matured in nascent small ribosomal particles. The last proteins to get stably incorporated into pre-40S particles are Rps10 and Rps26, which are located on the beak and the platform [9,44,45]. The wriggling here observed might reflect this late maturation process.

Second, rRNA chemical probing (ChemModSeq) experiments performed on yeast pre-40S particles revealed that nucleotides forming both the head and platform regions were more accessible to chemical probes and thus more structurally unstable in intermediate pre-40S particles, like the ones herein studied, than later pre-ribosomal particles [28]. This observation is consistent with the vibrating states that we described.

Third, instability or unfolding of the platform was already reported on other high-resolution cryo-EM structures of pre-40S and 80S-like particles obtained by our group [19] and others (Rai et al., Biorxiv doi: https://doi.org/10.1101/617910). Reproducibility of this observation somehow reduces the risk of punctual sample mishandling leading to structural distortions of the pre-40S particles of interest.

Of note, none of the structural classes we observed by cryo-EM and image analysis displayed a structurally stable beak together with a vibrating platform. Based on the assumption that vibration of the head and platform domains might be a true feature of maturing pre-40S particles, we proposed that the beak is first to enter into a vibrating state, while platform wriggling would be a subsequent event.

Our bottom-up proteomic data revealed that Tsr1-FPZ pre-40S particles were lacking Rps10 and Rps31, which are located on the beak of the mature 40S subunit, as well as Rps26 (on its platform). Furthermore, like in other published cytoplasmic early pre-40S cryo-EM maps, the 18S rRNA three-way junction formed by helices h34, h35 and h38 on the base of the head is in an immature conformation. We hypothesized that wriggling of the head region might be a reflection of rRNA maturation, as well as stable incorporation of “missing” RPs of the head (Rps3, Asc1, Rps10 and Rps31) of the small ribosomal subunit.

In wild type conditions, the final maturation steps of the pre-40S particle are thought to be the cleavage of the 18S rRNA 3’-end by Nob1 and the release of Dim2 and Nob1 from the platform. Dim2 release would allow the stable incorporation of Rps26 under the newly matured 18S rRNA 3′-end [13]. Here again, we speculated that platform vibration might be either a cause or a consequence of these last maturation events. The mechanisms that trigger platform wriggling remain to be established.

We combined our herein results with previous knowledge based on functional experiments, in order to propose a putative chronology of remodeling events occurring to yeast pre-40S particles when going from early to intermediate cytoplasmic maturation states. This model is presented in Figure 5, and is briefly discussed hereafter.

### 3.2. Dim1 Activity and Release from pre-40S Particles Might Correspond to Early Cytoplasmic Events

Dim1 was found in nucleolar pre-40S particles, but it dimethylates the 18S rRNA on A1780 and A1781 in the cytoplasm [18,46]. This raises the question of the precise timing of action and release of this protein from cytoplasmic pre-40S particles. Primer extension analyses have revealed the presence of these modifications on early and intermediate cytoplasmic pre-40S particles purified using Ltv1, Enp1 or Tsr1 as baits [28], indicating that rRNA dimethylation occurs early in the cytoplasm. Our bottom-up proteomic and cryo-EM analyses show that Dim1 is absent from the Tsr1-FPZ pre-40S particles that we purified. This might be caused by our sample preparation conditions, or by the fact that Dim1, having already performed its enzymatic activity, has already left Tsr1-FPZ pre-40S particles.

Consistently, cryo-EM analyses of yeast cytoplasmic pre-40S particles purified in wild-type conditions using Rio2 [15] or Ltv1 [16] as baits showed that Dim1 was only detected in a minor fraction of the analyzed population, suggesting the high lability of this protein. The C-terminal domain of Dim1 could only be unambiguously resolved in cryo-EM studies of pre-40S particles, which were purified in mutant conditions: the first one used a tagged and catalytically inactive version of Nob1 as bait [17], the second one used a tagged version of Tsr1 as bait (like herein), within a genetic background carrying a deletion of a flexible loop of Rps20 [19]. In this latter case, western blots suggested that this deletion prevented several RBFs including Dim1 to dissociate from maturing Tsr1-purified particles. There again, in both cryo-EM studies, Dim1 was only visible in a fraction of the particles, reinforcing the idea that this protein dissociates at an early step or is structurally labile within pre-40S particles. Biochemical as well as cryo-EM studies of later 80S-like particles purified in mutant strains [34] (Rai et al. BioRxiv https://doi.org/10.1101/617910) reinforce the idea that Dim1 is only retained on late pre-40S particles upon impairment of maturation.

Altogether, we postulated that in wild-type conditions, Dim1 dimethylation activity and subsequent dissociation are early cytoplasmic events, which might occur even before the release of Ltv1 from pre-40S particles. In our model, we thus placed Dim1 activity and subsequent release as one of the first cytoplasmic events (Figure 5, lower left panels).

### 3.3. Ltv1 Conformational Change and Beak Wriggling Precede Ltv1 Phosphorylation by Hrr25

Local classifications focused on the head domain revealed that Tsr1-FPZ pre-40S particles were divided into three different classes (H1-H3; see Figure 3 and Appendix A). H1 was the most structurally stable and could thus be auto-refined up to a resolution of 3.8 Å. In that class, association of Ltv1 and Enp1 was clearly visible on the beak, which presented well-resolved features, with its rRNA in an immature conformation. This class was identical to other existing early cytoplasmic pre-40S head structures, and was consequently attributed to an early cytoplasmic maturation step. H2 was refined to 6.4 Å and harbored a beak with more blurred features, suggesting a higher flexibility of this region compared to the first class. This lower resolution map, harboring a mild wriggling of the beak domain also displayed a conformational change of Ltv1. Whether the wriggling of the beak induces Ltv1 conformational change or the contrary remains to be elucidated. These two events are unlikely to be caused by Ltv1 and Enp1 phosphorylation by Hrr25. Indeed, firstly, our bottom-up proteomic analyses revealed very small amounts of phosphorylation of Ltv1 and Enp1 on S336/S339 and S172, respectively, (6.5% for Ltv1 and 2.1% for Enp1). These percentages are much lower than the number of pre-40S particles harboring a conformational change in Ltv1 position as well as beak destabilization (20% of particles in H2 state). Secondly, previous genetic and functional analyses revealed that the interaction between positively charged N-terminal residues of Rps3 and negatively charged amino acids of Rps20 triggers a conformational change in Ltv1, which might in turn help the recruitment of Hrr25; phosphorylation would then be quickly followed by the release of Ltv1 from the maturing particles [19]. Although we could not clearly detect the Rps3-Rps20 interaction in our cryo-EM maps, we hypothesized that the Ltv1 conformational change and beak mild wriggling that we observed was a direct result of this interaction, and that it preceded and favored Ltv1 and Enp1 phosphorylation by Hrr25. A third class of particles (H3) displayed heads with highly blurred features, and could not be refined to subnanometer resolutions, suggesting an even higher degree of dynamism than the two other structural classes. We postulated that this strong wriggling of the head might be due to the release of phosphorylated Ltv1 and Enp1 from the particles (Figure 5, upper right panels).

## 4. Material and Methods

### 4.1. Purification of Cytoplasmic pre-40S Particles

Pre-ribosomal particles were purified from a BY4742 *Saccharomyces cerevisiae* strain harboring a C-terminal tandem affinity purification tag on Tsr1 (Tsr1-FPZ). This FPZ tag is composed of a Flag peptide, a PreScission cleavage site and a Z domain derived from *S. aureus* protein A [35]. 12 L of yeast cells were grown at 30 °C on YPD medium (1% yeast extract, 1% peptone, supplemented with 2% glucose) to an optical density between 1.2 and 1.8 (600 nm), and harvested by centrifugation at 4000 rpm for 10 min at 4 °C. Pellets were washed twice with ddH2O, and flash frozen in liquid nitrogen. Frozen yeast cells were then lysed by cryogenic grinding, using a Planetary Ball Mill PM100 (Retsch, Verder Scientific, Haan, Germany).

The yeast grindate was then resuspended in ice-cold buffer A (50 mM Tris-HCl, pH 8, 150 mM NaCl, 10 mM MgCl_2_, 0.1% Igepal supplemented with Complete EDTA free protease inhibitor (Roche)), and cleared by centrifugation (4 °C, 15 min, 10,000 g). The supernatant was incubated on IgG Sepharose^TM^ Fast Flow beads (Fisher Scientific) for 1 h on a rocking table at 4 °C. Beads were washed with buffer B (20 mM HEPES, pH 7.6; 150 mM NaCl, 100 mM KOAc, 10 mM MgCl_2_, 1 mM DTT, 0.02% Tween 20, 0.1% Triton), and IgG-bound complexes were subsequently eluted by PreScission-Protease cleavage (2 h, 4 °C). The eluted solution was subjected to a second affinity purification step by incubation with Anti-Flag^®^ M2 Affinity Gel (Sigma-Aldrich) in buffer C (20 mM HEPES, pH 7.6, 150 mM NaCl, 100 mM KOAc, 10 mM MgCl_2,_ 0.02% Tween 20, 0.1% Triton) at 4 °C, immediately washed in buffer D (10 mM Tris-HCl, pH 7.4, 150 mM NaCl, 10 mM MgCl_2_) and eluted in the same buffer supplemented with 0.4 mg/mL 2X Flag peptide (custom-made, IGBMC, Strasbourg, France).

Flag eluates were layered on top of a 10%–30% sucrose gradient in buffer E (20 mM HEPES, pH7.6, 150 mM NaCl, 10 mM MgCl_2_, 1 mM DTT). Eleven milliliter-containing gradient tubes were submitted to ultracentrifugation on a Beckman Coulter Optima L-100 XP Ultracentrifuge, SW41 rotor (3 h 20 min, 38,000 rpm, 4 °C), and gradient fractions were subsequently scanned at A = 254 nm and collected using a Fraction Collector system (Foxy R1, Teledyne Isco). Fractions of interest were pooled and sucrose was removed by 5 successive series of concentrating/washing on Vivacon 2 100 kDa MWCO centrifugal devices (Sartorius, Göttingen, Germany).

### 4.2. Protein Digestion and NanoLC-MS/MS Analysis

Twenty five microliters of concentrated Tsr1-FPZ pre-40S particles, corresponding to 5 µg of protein were reduced by incubation for 5 min at 95 °C with 5 µL of Laemmli buffer containing 25 mM DTT, then alkylated with 100 mM iodoacetamide for 30 min at room temperature in the dark. Samples were then loaded and concentrated on a SDS-PAGE gel. For this purpose, the electrophoresis was stopped as soon as the proteins left the stacking gel to enter the resolving gel as one single band. The proteins, revealed with Instant Blue (Merck KGaA, Darmstadt, Germany) for 20 min, were found in one blue band of around 5 mm width. The band was cut and washed before the in-gel digestion of the proteins overnight at 37 °C with a solution of modified trypsin. The resulting peptides were extracted from the gel using two successive incubations in 10% formic acid (FA)/acetonitrile (ACN) (1:1, *v*/*v*) for 15 min at 37 °C. The two collected fractions were pooled, dried and resuspended with 25 μL of 2% ACN, 0.05% trifluoroacetic acid (TFA). NanoLC-MS/MS analysis was performed in duplicate injections using an Ultimate 3000 nanoRS system (Thermo Fisher Scientific, Bremen, Germany) coupled to an LTQ-Orbitrap Velos mass spectrometer (Thermo Fisher Scientific, Bremen, Germany) operating in positive mode. Five microliters of each sample was loaded onto a C18-precolumn (300 μm inner diameter × 5 mm) at 20 μL/min in 2% ACN, 0.05% TFA. After 5 min of desalting, the precolumn was switched online with the analytical C18 nanocolumn (75 μm inner diameter × 15 cm, packed in-house) equilibrated in 95% solvent A (5% ACN, 0.2% FA) and 5% solvent B (80% ACN, 0.2% FA). Peptides were eluted by using a 5%–25% gradient of solvent B for 75 min, then a 25%–50% of solvent B for 30 min at a flow rate of 300 nL/min. The LTQ-Orbitrap Velos was operated in data-dependent acquisition mode with the XCalibur software (Thermo Fisher Scientific, Bremen, Germany). Survey scans MS were acquired in the Orbitrap, on the 350–1800 *m*/*z* (mass to charge ratio) range, with the resolution set to a value of 60,000 at *m*/*z* 400. Up to twenty of the most intense multiply charged ions (2+ and 3+) per survey scan were selected for CID fragmentation, and the resulting fragments were analyzed in the linear ion trap (LTQ). Dynamic exclusion was used within 60 s to prevent repetitive selection of the same peptide.

### 4.3. Bioinformatic MS Data Analysis

The Mascot (Mascot server v2.6.1; http://www.matrixscience.com) database search engine was used for peptide and protein identification using the automatic decoy database search to calculate a false discovery rate (FDR). MS/MS spectra were compared to the UniProt *S. cerevisiae* database. Mass tolerance for MS and MS/MS was set at 5 ppm and 0.8 Da, respectively. The enzyme selectivity was set to full trypsin with two missed cleavages allowed. Protein modifications were fixed carbamidomethylation of cysteines, variable phosphorylation of serine and threonine, variable oxidation of methionine and variable acetylation of protein N-terminus. Proline software was used for the validation and the label-free quantification of identified proteins in each sample (http://proline.profiproteomics.fr/) [47]. Mascot identification results were imported into Proline. Search results were validated with a peptide rank = 1 and at 1% FDR both at the PSM level (on the adjusted e-value criterion) and protein sets level (on the modified mudpit score criterion). Label-free quantification was performed for all proteins identified: peptides were quantified by extraction of MS signals in the corresponding raw files, and post-processing steps were applied to filter, normalize and compute protein abundances. Peptide intensities were summarized in protein abundance values using the median function.

### 4.4. Grid Preparation and cryo-EM Images Acquisition

Cryo-EM grids were prepared and systematically checked at METI, Toulouse. Immediately after glow discharge, 3.5 µL of purified pre-40S particles (with RNA concentrations of 35 ng/µL as estimated by the Nanodrop measurement) were deposited onto QUANTIFOIL^®^ holey carbon grids (R2/1, 300 mesh with a 2 nm continuous layer of carbon on top). Grids were plunge-frozen using a Leica EM-GP automat (Leica Camera AG; Wetzlar, Germany); temperature and humidity level of the loading chamber were maintained at 20 °C and 95% respectively. Excess of solution was blotted with a Whatman filter paper no.1 for 1.7–1.9 sec and grids were immediately plunged into liquid ethane (−183 °C).

Images were recorded on the CM01 beamline at the European Synchrotron Radiation Facility (ESRF), Grenoble. The CM01 Titan Krios electron microscope (Thermo Fisher Scientific, Bremen, Germany) was operating at 300 kV and was equipped with a Gatan K2 summit direct electron detector using counting mode. Automatic image acquisition was performed with EPU, at a magnification corresponding to a calibrated pixel size of 1.067 Å and a total electron dose of 29.4 e^−^/Å^2^ over 28 frames. Nominal defocus values ranged from −0.8 to −2.8 µm.

### 4.5. Single Particle Analysis

Three thousand and one hundred stacks of frames were collected at ESRF. Frames stacks were aligned to correct for beam-induced motion using MOTIONCORR2 [48]. Contrast Transfer Function (CTF) and defocus estimation was performed on the realigned stacks using CTFFIND4 [49]. After selection upon CTF estimation quality, maximum resolution on their power spectra and visual checking, “good” micrographs were retained for further analysis. 221,234 particles were automatically picked using the RELION 2.1 Autopick option, and then were extracted in boxes of 384 pixels × 384 pixels. All subsequent image analysis was performed using RELION 2.1 [50] (Appendix A). A first 2D classification was performed (on particles images binned by a factor of 8) to sort out ill-picked particles. The 144,890 remaining particles were binned by a factor of 4 and subjected to a 3D classification in 6 classes, using the 40S subunit extracted from the crystal structure of *S. cerevisiae* full ribosome (PDB-ID 4V88) [51], low-pass filtered to 60 Å, as initial reference. Two classes harbored full 40S morphology and good level of details. Thus, particles from these classes were grouped and re-extracted without imposing any binning factor, and a consensus 3D structure was obtained using RELION’s 3D auto-refine option, with an overall resolution of 3.1 Å for FSC = 0.143 according to gold-standard FSC procedure (Appendix A).

As the head and platform domains harbored highly blurred and thus structurally heterogeneous features, the 74,769 particles of the consensus reconstruction were first submitted to a global 3D classification scheme, which yielded 3 3D classes, GC1-GC3, that were further auto-refined and post-processed up to resolutions of 3.8, 3.2 and 3.7 Å, respectively (Appendix A). In parallel, structural heterogeneity of the head and platform regions was assessed using focused 3D classifications with signal subtraction [52] around the pre-40S particles head and platform (Appendix A). For all reconstructions auto-refined to 9 Å and higher, detector MTF correction and global map sharpening and was performed using RELION’s Postprocessing option, with an automatic B-factor estimation (Appendix A).

### 4.6. Cryo-EM Maps Interpretation

Atomic models of pre-40S particles (PDB-ID 6EML and PDB-ID 6FAI) [16,17] were first fitted in the cryo-EM maps of interest as rigid body using the “fit” command in UCSF Chimera [53]. Manual refinements and adjustments, as well as flexible and jiggle fittings were then realized on various chains of the models in Coot [54].

Final atomic model of early cytoplasmic pre-40S particle was refined using REFMAC5 [55] and Phenix_RealSpace_Refine [56], with secondary structure restraints for proteins and RNA generated by ProSMART [57] and LIBG [54]. Final model evaluation was done with MolProbity [58]. Overfitting statistics were calculated by a random displacement of atoms in the model, followed by a refinement against one of the half-maps in REFMAC5, and Fourier shell correlation curves were calculated between the volume from the atomic model and each of the half-maps in REFMAC5 (Appendix A).

Maps and models visualization was done with Coot and UCSF Chimera; figures were created using UCSF Chimera.

## Figures and Tables

**Figure 1 molecules-25-01125-f001:**
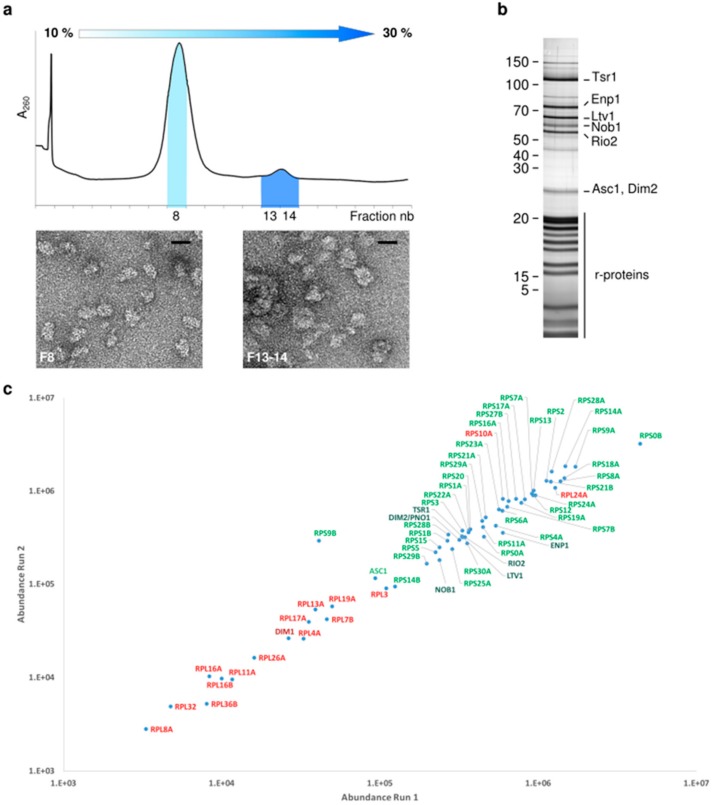
Purification of Tsr1-FPZ pre-40S particles. (**a**) Sucrose gradient profile of eluted Tsr1-FPZ pre-40S particles (upper panel) and negative staining TEM observations of gradient fractions 8 and 13–14 (lower panels). Scale bars represent 25 nm. (**b**) Silver stained SDS-PAGE gel of sucrose gradient fraction 8, corresponding to Tsr1-FPZ pre-40S particles. (**c**) Label-free bottom-up proteomic analyses of fraction 8. Proteins marked in green represent ribosomal proteins (light green) or RBFs (dark green) with an observed/observable peptide ratio > 30%, while proteins indicated in red have a lower ratio.

**Figure 2 molecules-25-01125-f002:**
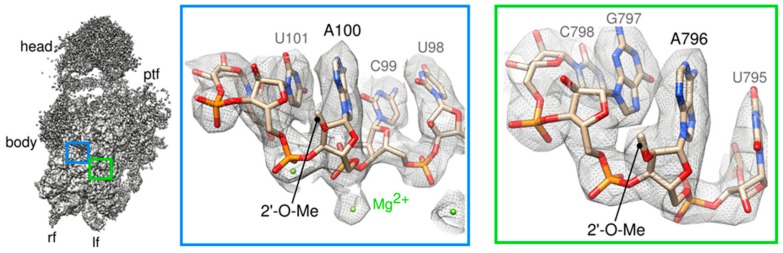
Consensus cryo-EM map of Tsr1-FPZ pre-40S particles. Left panel represents a view from the intersubunit side of the consensus cryo-EM map, solved to 3.1 Å resolution. Regions of the map located on the body of the pre-40S particle (contoured by blue and green rectangles) that display electron density corresponding to known nucleotide modifications are zoomed in on the middle and right panels, respectively.

**Figure 3 molecules-25-01125-f003:**
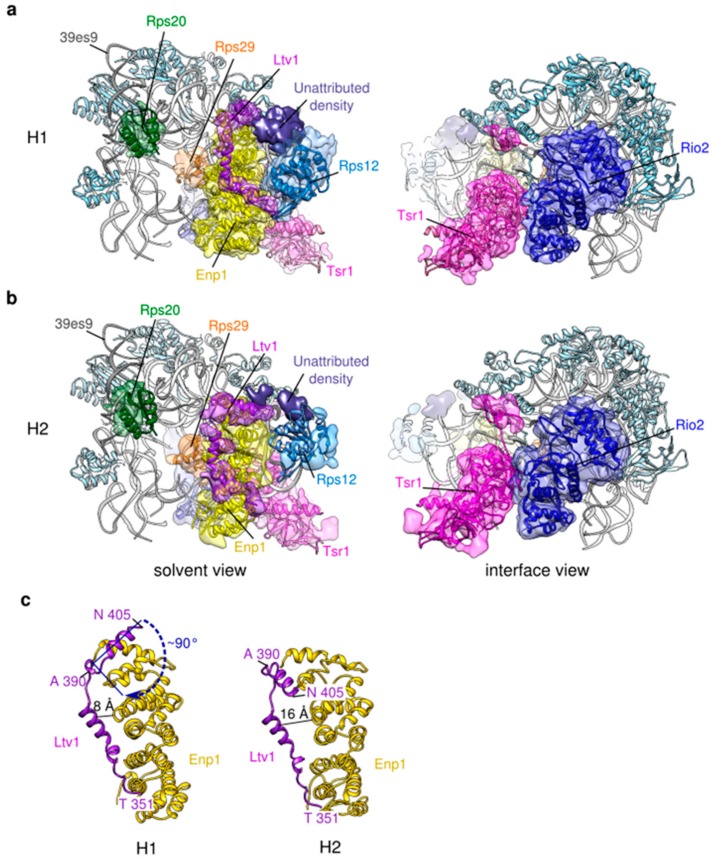
Focused 3D classification and refinement of the head domain of Tsr1-FPZ pre-40S particles. (**a**) Structural details of 3D class H1, refined up to 3.8 Å resolution. Only parts of the cryo-EM map corresponding to Rps20 (dark green), Rps29 (orange), Enp1 (yellow), Ltv1 (purple), Rps12 (light blue), Rio2 (dark blue) and Tsr1 (magenta) and to an unattributed protein have been segmented and displayed. Other ribosomal proteins of the beak regions are shown by light blue ribbons, while rRNA is shown in silver. (**b**) Structural details of 3D class H2, solved to 6.4 Å resolution. Color codes and map segmentations are identical to those used for H1. Solvent views are shown on the left panels, while intersubunit views are on the right panels for (**a**,**b**). (**c**) comparison of the structure position of Ltv1 and Enp1 in H1 (left) and H2 (right).

**Figure 4 molecules-25-01125-f004:**
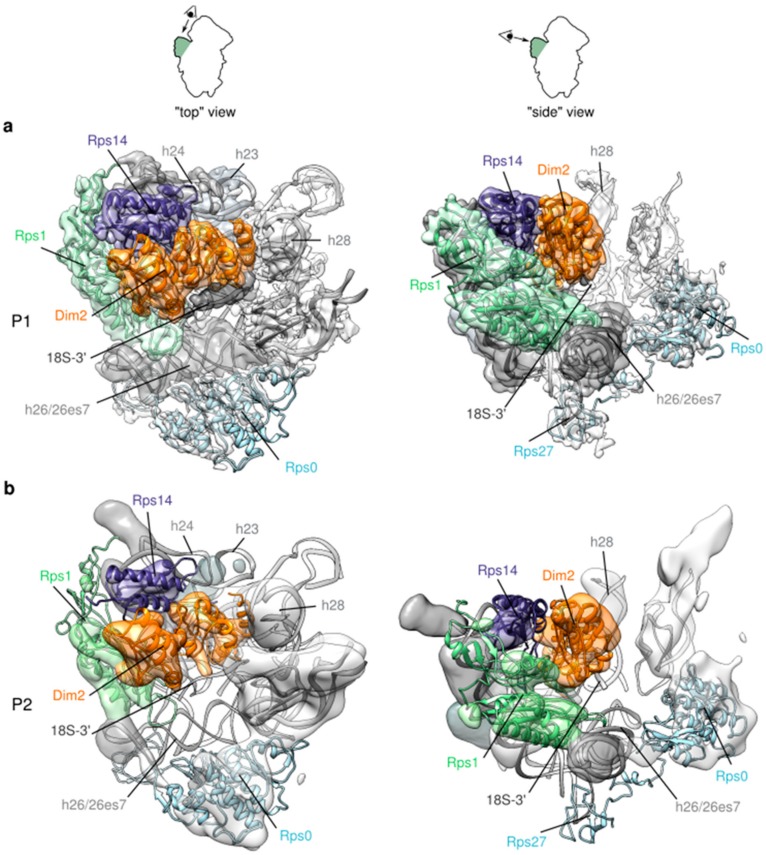
Focused 3D classification and refinement of the platform domain of Tsr1-FPZ pre-40S particles. (**a**) Structural details of 3D class P1, solved to 3.7 Å resolution. The full cryo-EM map of this region is displayed. Rps1 is shown in light green, Rps14 in violet, Dim2 in orange. Other ribosomal proteins are shown in light blue ribbons in white electron densities, while rRNA is shown in light grey, with the exception of its 3′-end region, displayed in dark grey. (**b**) Structural details of 3D class P2, refined to a resolution of 9.1 Å. Like for P1, all the electron densities of the cryo-EM map are displayed and color codes are identical to P1.

**Figure 5 molecules-25-01125-f005:**
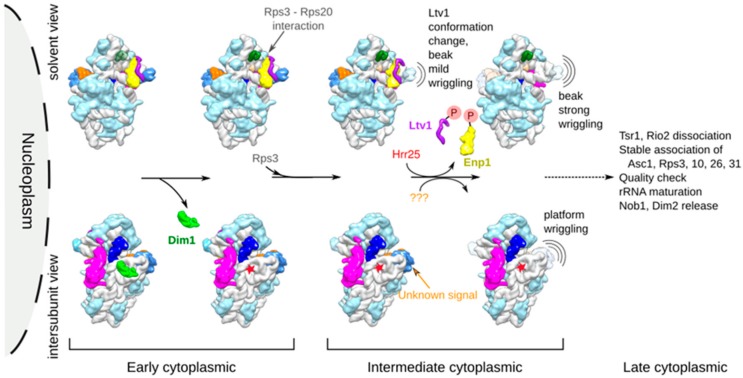
Chronological model of structural remodeling events occurring to pre-40S particles during early and intermediate cytoplasmic maturation stages. Upper panel shows events occurring on the solvent side of pre-40S particles, while lower panels represents events of the intersubunit side. Red star indicates base dimethylation of A1781 and A1782. Orange question marks (???) represent the unknown signal triggering platform wriggling.

**Table 1 molecules-25-01125-t001:** Relative abundance of phosphorylation sites detected by bottom-up proteomics on Tsr1-FPZ pre-40S particles components.

Protein	Peptide Sequence	Phosphorylation Protein Position	% Phosphosite run 1	% Phosphosite run 2	% Phosphosite Average	
Tsr1	DTLDEYAPEGTEDWSDYDEDFEYDGLTTAR	T354 or S358	100	100	0	Non phosphorylated
					100	Phosphorylated
Ltv1	GAMSDVSGFSMSSSAIAR	S336 or S339	7.8	5.2	93.5	Non phosphorylated
					6.5	Phosphorylated
Ltv1	VTNTLSSLKF	S460	0	2.5	98.7	Non phosphorylated
					1.3	Phosphorylated
Enp1	EKESQVEDMQDDEPLANEQNTSR	S172	0	4.3	97.9	Non phosphorylated
					2.1	Phosphorylated
Enp1	ESQVEDMQDDEPLANEQNTSRGNISSGLK	T189 or S195 or S196	0	1.2	99.4	Non phosphorylated
					0.6	Phosphorylated
Enp1	ILDDGSNGEDATR	S404	0.2	0.2	99.8	Non phosphorylated
					0.2	Phosphorylated

## Data Availability

Mass spectrometry proteomics data have been deposited to the ProteomeXchange Consortium via the PRIDE [59] partner repository with the dataset identifier PXD016577. Cryo-EM maps have been deposited in the Electron Microscopy Data Bank (EMDB), under the accession codes: EMD-10715 (Consensus Tsr1-FPZ); EMD-10713 (GC1 Tsr1-FPZ); EMD-10716 (H1 Tsr1-FPZ); EMD-10717 (H2 Tsr1-FPZ); EMD-10718 (P1 Tsr1-FPZ); EMD-10719 (P2 Tsr1-FPZ). Atomic coordinate model of an early cytoplasmic Tsr1-FPZ pre-40S particle has been deposited in the Protein Data Bank (PDB), with accession code PDB: 6Y7C.

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
