# Peer review of "Good Vibrations: Structural Remodeling of Maturing Yeast Pre-40S Ribosomal Particles Followed by Cryo-Electron Microscopy"

_molecules, 2020, doi:10.3390/molecules25051125_

Round 1

Reviewer 1 Report

The manuscript from Shayan et al. explors the structural variability in a sample of the small ribosomal subunit arrested at a late maturation step. The methodology used is clearly explained and adequate. The results are well written and technically sound. They are discussed in the context of previous results and a chronological model of the structural changes occurring during maturation of the small ribosomal subunit in the cytoplasm is proposed.

The analysis of the cryo-EM dataset is well done and the authors characterized different states as it is often the case for cryo-EM analysis of multiprotein complexes. Some of the states reached less than 4 Å resolution and the interpretation of these states is straightforward. Then, the authors characterized as well some much more flexible states where some proteins present in the stable states are missing and their interpretation is more challenging. Nevertheless, they inferred that this flexibility plays a role for the subsequent steps of the maturation pathway of the small ribosomal subunit.

Overall, the interpretation of the results looks fine but the authors shouldn’t over interpret their results as well. How can the authors be sure that the wriggling of the beak plays a role in the maturation pathway? This wriggling could as well result from heterogeneity in the sample caused by the various steps of purification, preparation of cryo-EM grid ...etc... It should be clarified.

Line 128-129: To rephrase:

"Label-free quantification the relative abundances of the identified proteins
showed that these pre-40S were composed of all RPS with the exception of Rps26 and Rps31."

Supplementary Table1: Report the rotamer statistics from molprobity

Author Response

(Reviewer's comments are in black, authors' reply are in blue, manuscript modifications are in red).

The manuscript from Shayan et al. explors the structural variability in a sample of the small ribosomal subunit arrested at a late maturation step. The methodology used is clearly explained and adequate. The results are well written and technically sound. They are discussed in the context of previous results and a chronological model of the structural changes occurring during maturation of the small ribosomal subunit in the cytoplasm is proposed.

The analysis of the cryo-EM dataset is well done and the authors characterized different states as it is often the case for cryo-EM analysis of multiprotein complexes. Some of the states reached less than 4 Å resolution and the interpretation of these states is straightforward. Then, the authors characterized as well some much more flexible states where some proteins present in the stable states are missing and their interpretation is more challenging. Nevertheless, they inferred that this flexibility plays a role for the subsequent steps of the maturation pathway of the small ribosomal subunit.

Overall, the interpretation of the results looks fine but the authors shouldn’t over interpret their results as well. How can the authors be sure that the wriggling of the beak plays a role in the maturation pathway? This wriggling could as well result from heterogeneity in the sample caused by the various steps of purification, preparation of cryo-EM grid ...etc... It should be clarified.

We are very thankful for the positive appreciation of our work by Reviewer 1. We do agree that results overinterpretation must be avoided by all means. Furthermore, all reviewers are right to say that we can not formally rule out that the wriggling of the head and platform domains that we observe might be an artifact due to sample preparation and/or mishandling. We have thus added this notion in all three sections (introduction, results and discussion) of our manuscript (for the full modified text, see red sentences in the new version of the manuscript). For instance, at the end of paragraph 2.3 of the results, we have added the following sentences : “Although we cannot rule out that artifacts due to sample preparation or handling could lead to destabilization or destruction of the head domain, our observations might also indicate that this region is indeed more dynamic than the body. As instability of the beak domain was also previously observed by RNA chemical probing (ChemModSeq) studies [28], we speculate that after a first (early) step where it is structurally stable, the head domain of cytoplasmic pre-40S particles enters a more unstable, “wriggling” stage, that might be a true structural feature of intermediate pre-40S particles.”

The end of Results, 2.4 (focused classifications of the platform region) was also modified as follows: There again, platform instability might result from purification and/or cryo-EM grid preparation conditions. However, because vibration or unfolding of the platform has already been observed on various precursors to the small ribosomal subunit [19, 28, Rai et al. https://doi.org/10.1101/617910], we make the hypothesis that like for the head domain, after a maturation step where the platform is structurally stable, a signal of an unknown nature could trigger its wriggling. This structural instability of the platform might reflect the final steps of cytoplasmic pre-40S particles maturation (see discussion below).”

The end of the introduction (in which we exposed our findings) was also somehow toned down, by adding the following lines: “Our results indicate that particles can be subdivided into several structural classes, all of them harboring an almost mature body. The first one, overall structurally stable, likely correspond to an early maturation state. Other pre-40S particles, that we attribute to intermediate maturation states, would see first their beak and then their platform enter a highly dynamic (“vibrating” or “wriggling”) state. Based on these observations and previously published results, we hypothesize that structural instability of these domains might be a reflect of the remodeling events associated with correct maturation of 20S pre-rRNA and final RPS assembly. These structural results reinforce the idea that RBFs, besides their putative enzymatic activities, help stabilizing pre-rRNA into an immature conformation, thus preventing incompetent pre-40S particles to enter into the translating pool of small ribosomal subunits.”

In an attempt to better justify why we think the beak and platform wriggling that we observe might not be an artifact, we substantially modified the discussion part, and tried to better connect our structural observations to already existing litterature suggesting structural instability of the head and platform along pre-40S maturation. We have also tried to simplify this section, for instance by removing the part regarding the order of wriggling of the beak and platform, and integrating this notion within the very first paragraph of the discussion. Finally, the last paragraph regarding the potential role of Fap7 on platform maturation events, which was purely speculative, was removed, and Figure 5 was modified accordingly. We hope these modifications will convince you that this manuscript in worth publishing in this journal.

Line 128-129: To rephrase:

"Label-free quantification the relative abundances of the identified proteins
showed that these pre-40S were composed of all RPS with the exception of Rps26 and Rps31."

The sentence was modified as follows : « Label-free quantification of the relative abundances of the identified proteins indicated that these pre-40S were composed of all RPS with the exception of Rps26 and Rps31 ». 

Supplementary Table1: Report the rotamer statistics from molprobity

The percentage of rotamer outliers (1.0%) was added to this table

Reviewer 2 Report

This manuscript reports the heterogeneous structure of a pre-40S ribosomal intermediate. This precursor to the small ribosomal subunit is purportedly the cytoplasmic intermediate bound to the assembly factors Tsr1, which was the bait for the pulldown, as well as Rio2, Ltv1, and Dim2. Dim1 and Nob1 were not identified in the mass spectrometry data or the structure.

The core structure is ordered at below 4 Å resolution, allowing identification of methylated adenosines, but there is significant degradation of the head domain. Local classification/refinement reveals higher resolution at the head and platform in some subset of the particles. This is not atypical behavior for these precursor ribosomal intermediates, but interpreting the biological relevance of these observations is tricky because it is unclear whether the “wriggle” that the authors describe is really part of the maturation mechanism or if it is a consequence of the buffer environment after blotting/plunging into liquid ethane. The SSU is known to adopt a preferred orientation, so there is almost certainly an interaction between the molecule described here and the air-water interface of the blotted grid.  If this is why the head is unraveled, then interpreting the unraveled molecules as biologically relevant for folding is an overinterpretation of an artifactual result. The complete absence of Dim1 from their molecules is also curious as others have purified a similar intermediate by use of a catalytically inactive Nob1-TAP tag that retains Dim1 in at least a fraction of the particles (see Scaiola, et al). Dim1 should not fully dissociate until after 20S cleavage by Nob1. This suggests that the observed unfolding could be an artifact resulting from a damaged particle.

The trick is how to test whether this wriggle is something that happens in the cell or not. That is, of course, not an easy task. Reconstitution with the remainder of the Rps proteins (Rps3, 10, and 26) to see if the head then zips up into its mature conformation would show that this unfolded head/platform is functional and not an artifact. The structure need not be determined at atomic resolution to demonstrate folding of the head so hopefully this control is not too onerous. Alternatively, multibody refinement in RELION, analyzing the head independently from the body and/or the beak, platform, and body as three independent bodies might also illuminate this structural mobility by independently refining the folding domains within the same structure, rather than as subpopulations of the whole. That would show that the blurring of the structure is from discrete movements of the head, as it does in the mature 40S, rather than unfolded portions of the structure.

There were some instances of curious grammar/word choice that might be reconsidered by the authors, for example the use of the word vibratile/vibratility (I had to look it up!). On line 120 on page 3, “A smaller peak on” should read “A smaller peak of”. The referenced biorx manuscript is Rai, et al (not Jai et al). Page 6, line 174, “adenosin” should be “adenosine”.

Author Response

(Reviewer's comments are in black, authors' reply are in blue, manuscript modifications are in red).

This manuscript reports the heterogeneous structure of a pre-40S ribosomal intermediate. This precursor to the small ribosomal subunit is purportedly the cytoplasmic intermediate bound to the assembly factors Tsr1, which was the bait for the pulldown, as well as Rio2, Ltv1, and Dim2. Dim1 and Nob1 were not identified in the mass spectrometry data or the structure.

We would like to thank referee 2 for reviewing our manuscript and giving these constructive remarks. However, we would like to draw the attention of referee 2 on the following point : as written in the first paragraph of the results section and shown in Figure 1c, Nob1 was indeed found in the BU proteomics data. However, like in all other published structures of yeast pre-40S particles, we could not clearly distinguish it on our cryo-EM maps. Dim1 is the only factor that we could not detect either by proteomics or cryo-EM.  

The core structure is ordered at below 4 Å resolution, allowing identification of methylated adenosines, but there is significant degradation of the head domain. Local classification/refinement reveals higher resolution at the head and platform in some subset of the particles. This is not atypical behavior for these precursor ribosomal intermediates, but interpreting the biological relevance of these observations is tricky because it is unclear whether the “wriggle” that the authors describe is really part of the maturation mechanism or if it is a consequence of the buffer environment after blotting/plunging into liquid ethane. The SSU is known to adopt a preferred orientation, so there is almost certainly an interaction between the molecule described here and the air-water interface of the blotted grid.  If this is why the head is unraveled, then interpreting the unraveled molecules as biologically relevant for folding is an overinterpretation of an artifactual result.

We agree with Referee 2, as well as the two other referees, that we cannot rule out that the wriggling of the head and platform that we observe might be an artifact due to sample preparation and/or mishandling. We have thus tried to add this notion in all three main chapters (intro, results, discussion) of our manuscript (see red sentences in the newest version of the article, and some examples below). We do observe a preferential orientation of the Tsr1-FPZ pre-40S particles on the cryo-EM grids: particles tend to lie flat or on the side within the ice layer, thus head/top and feet/bottom views are scarce (see Suppl. Figure1) . If there was some destruction of the structure arising from interactions with the air/water interface, because of the orientation of the sample within the ice layer, we would expect it to reach other domains of the pre-40S particles including regions of the body, and not only of the head and platform. In the particles that we studied, the body is always extremely well resolved. Thus, we do not believe that the vibration of the head and platform that we observe is caused by preferential orientation within the ice layer.

Published results from other labs or our own led us to believe that this vibration might be a true event occurring to intermediate pre-40S particles : as stated in the manuscript (see the first paragraph of the revised discussion), several studies performed in vitro on prokaryotic SSU (Nomura, 1991) or by systematic RPS deletions in eukaryotic cells (Ferreira-Cerca et al., 2007; O’Donohue et al, 2010) suggested that the rRNA 3’ major domain (corresponding to the beak and platform domains) might be the last part of small ribosomal subunits to be assembled. Later on, RNA probing experiments performed by the group of S. Granneman (Hector et al., 2014) revealed a higher dynamism of this 3’ major domain of pre-40S particles purified using Rio2, Tsr1, Ltv1 and Enp1 compared to later pre-40S particles lacking these ribosome biogenesis factors. Furthermore, cryo-EM experiments performed by our group (Mitterer, Shayan et al., 2019) and others (Rai et al. Biorxiv) reported the vibration/unfolding of the platform domain when purifying pre-40S particles in mutant conditions.

We understand nevertheless that this interpretation remains speculative and should be considered as such. We have revised the manuscript as follows :

- The end of the introduction (where our findings were previously briefly exposed) was somehow toned down, by adding the following lines: “Our results indicate that particles can be subdivided into several structural classes, all of them harboring an almost mature body. The first one, overall structurally stable, likely correspond to an early maturation state. Other pre-40S particles, that we attribute to intermediate maturation states, would see first their beak and then their platform enter a highly dynamic (“vibrating” or “wriggling”) state. Based on these observations and previously published results, we hypothesize that structural instability of these domains might be a reflect of the remodeling events associated with correct maturation of 20S pre-rRNA and final RPS assembly. These structural results reinforce the idea that RBFs, besides their putative enzymatic activities, help stabilizing pre-rRNA into an immature conformation, thus preventing incompetent pre-40S particles to enter into the translating pool of small ribosomal subunits.”.

- In the results section, several sentences were added to counterbalance our hypothesis in which beak and platform wriggling might be a true reflect of in vivo maturation events. For instance, at the end of results 2.2 sections, we added “Although we cannot rule out that artifacts due to sample preparation or handling could lead to destabilization or destruction of the head domain, our observations might also indicate that this region is indeed more dynamic than the body. As instability of the beak domain was also previously observed by RNA chemical probing (ChemModSeq) studies [28], we speculate that after a first (early) step where it is structurally stable, the head domain of cytoplasmic pre-40S particles enters a more unstable, “wriggling” stage, that might be a true structural feature of intermediate pre-40S particles.” The end of Results, 2.4 (focused classifications of the platform region) was also modified by adding : There again, platform instability might result from purification and/or cryo-EM grid preparation conditions. However, because vibration or unfolding of the platform has already been observed on various precursors to the small ribosomal subunit [19, 28, Rai et al. https://doi.org/10.1101/617910], we make the hypothesis that like for the head domain, after a maturation step where the platform is structurally stable, a signal of an unknown nature could trigger its wriggling. This structural instability of the platform might reflect the final steps of cytoplasmic pre-40S particles maturation (see discussion below).”(for the full modified text, see red sentences in the new version of the manuscript)

We also substantially modified the discussion, and tried to establish more convincing links between our structural observations and already existing litterature suggesting structural instability of the head and platform along pre-40S maturation. We have also tried to simplify this section, for instance by removing the part regarding the order of wriggling of the beak and platform, and integrating this notion within the very first paragraph of the discussion. The last paragraph of this section regarding the potential role of Fap7 on platform maturation events, which was purely speculative, was removed, and Figure 5 was modified accordingly.

The complete absence of Dim1 from their molecules is also curious as others have purified a similar intermediate by use of a catalytically inactive Nob1-TAP tag that retains Dim1 in at least a fraction of the particles (see Scaiola, et al).

Here, we would like to point out that the particles under the scope of our study are purified using a tagged version of Tsr1, within a wild type genetic background. Furthermore, as stated in the manuscript, we only studied pre-40S, not later 80S-like particles, which we excluded from our structural study by separating purified pre-ribosomal particles using sucrose gradient fractionation. This is not the case in Scaiola et al, where pre-40S and 80S-like particles were accumulated and purified using a tagged an inactived version of Nob1. Pre-40S and 80S-like particles were then sorted out on cryo-EM images of the prep, using single particle analysis methods. Thus, we disagree with the idea that we purified a similar intermediate than the one in Scaiola et al.

We believe it is important to distinguish wild-type from mutant conditions, in which some RBFs might be directly or indirectly retained on pre-40S particles because of the mutation. Therefore, we have now modified the discussion part of the manuscript as follows: “Our bottom-up proteomic and cryo-EM analyses show that Dim1 is absent from the Tsr1-FPZ pre-40S particles that we purified. This might be caused by our sample preparation conditions, or by the fact that Dim1, having already performed its enzymatic activity would have left Tsr1-FPZ pre-40S particles.

Consistently, cryo-EM analyses of yeast cytoplasmic pre-40S particles purified in wild-type conditions using Rio2 [15] or Ltv1 [16] as baits showed that Dim1 was only detected in a minor fraction of the analyzed population, suggesting the high lability of this protein. The C-terminal domain of Dim1 could only be unambiguously resolved in cryo-EM studies of pre-40S particles which were purified in mutant conditions: the first one used a tagged and catalytically inactive version of Nob1 as bait [17], the second one used a tagged version of Tsr1 as bait (like herein), within a genetic background carrying a deletion of a flexible loop of Rps20. Western blots suggested that this deletion prevented several RBFs including Dim1 to dissociate from maturing Tsr1-purified particles [19]. There again, in both cryo-EM studies, Dim1 was only visible in a fraction of the particles, reinforcing the idea that this protein dissociates at an early step or is structurally labile within pre-40S particles. Biochemical as well as cryo-EM studies of later 80S-like particles purified in mutant strains [34, Rai et al. BioRxiv https://doi.org/10.1101/617910] reinforce the idea that Dim1 is only retained on late pre-40S particles upon impairment of maturation.

Altogether, we postulate that in wild-type conditions, Dim1 dimethylation activity and subsequent dissociation are early cytoplasmic events, that might occur even before the release of Ltv1 from pre-40S particles. In our model, we have thus placed Dim1 activity and subsequent release as one of the first cytoplasmic events (Figure 5, lower left panels).

Dim1 should not fully dissociate until after 20S cleavage by Nob1. This suggests that the observed unfolding could be an artifact resulting from a damaged particle.

To our knowledge, there is no formal evidence that in wild-type conditions, Dim1 should fully dissociate until after 20S cleavage by Nob1 ; the last RBFs associated to either pre-40S or 80S-like particles are Rio1, Nob1 and Dim2 (Turowski et al., 2014 ; Hector et al., 2014).

We agree with Referee 2 that absence of Dim1 on the analyzed pre-40S particle pool might be partly due to harsh purification conditions, and we modified our manuscript to explicit this idea both in the result and discussion parts. For instance, at the end of the first chapter of the results sections, we added the following sentence : “Ribosome biogenesis factors Tsr1, Rio2, Enp1, Ltv1, Dim2 and Nob1 were found associated to pre-40S particles; however, only traces of Dim1 were found in the pool of Tsr1-FPZ pre-40S particles. We do not exclude that our purification protocol might artificially dissociate Dim1 from pre-40S particles, but other studies have suggested that this RBF has already left pre-ribosomes at this maturation stage (see below).”

However, since unfolding of the platform was previously observed both on pre-40S particles (Scaiola et al., 2018 ; Mitterer et al., 2019) and 80S-like particles (Rai et al., biorxiv) where Dim1 was retained by mutant conditions, we disagree with the idea that observed wriggling/unfolding could be an artifact resulting from a damaged particle triggered by the absence of Dim1.

The trick is how to test whether this wriggle is something that happens in the cell or not. That is, of course, not an easy task. Reconstitution with the remainder of the Rps proteins (Rps3, 10, and 26) to see if the head then zips up into its mature conformation would show that this unfolded head/platform is functional and not an artifact. The structure need not be determined at atomic resolution to demonstrate folding of the head so hopefully this control is not too onerous.

We agree with referee 2 that this would be a very elegant experiment to try, but even without aiming at atomic resolution, obtaining cryo-EM structures of reconstituted 40S subunits from pre-40S particles supplemented with Rps3, 10 and 26 (as well as at least Rio1 and maybe Fap7) might take much more than the time we have for the revision of this manuscript. Thus we did not undertake this experiment.

Alternatively, multibody refinement in RELION, analyzing the head independently from the body and/or the beak, platform, and body as three independent bodies might also illuminate this structural mobility by independently refining the folding domains within the same structure, rather than as subpopulations of the whole. That would show that the blurring of the structure is from discrete movements of the head, as it does in the mature 40S, rather than unfolded portions of the structure.

RELION’s multibody refinement is based on focused classifications followed by signal subtraction of designated domains of flexible particles, which is basically what we did for the head and platform domains of the Tsr1-FPZ pre-40S particles under the scope of this study. Multibody refinement will indeed help to describe the motions of a relatively well-structured domain compared to another. However it will not be able to better resolve the 3D structure of a domain that is intrinsically instable/wriggling than the approach that we used in this study (Nakane et al., eLife 2018).

Our strategy suggested that all the particles that we analyzed harbored a structurally stable and well resolved body. Within these, subpopulations of particles harbored either a structurally stable head (H1) or platform (P1), and this was confirmed by global classifications without masking or signal subtraction (see GC1). Other classes revealed intrinsic instability or wriggling : (H2, H3, P2, P3 for focused classification with signal subtraction, or GC2 and GC3 for global classifications). We believe that our analyses already reveal that the blurring of the head and platform in the consensus structure could originate both from discrete movements of structurally stable domains (H1, P1) and from a fraction of the particles population that would have become gradually instable/wriggling on their head (H2, H3) and platform domains (P2, P3).

Because of the relatively low resolutions of H2 (~6A) and P2 (~9A) and the impossibility to auto-refine H3 and P3 at resolution higher than 20 A (see manuscript), and the fact that all these structures were already obtained after focused classifications with signal subtraction, we are convinced that multibody refinement of the particles belonging to these structural classes would not yield better results than the ones presented here. The only interest of performing multibody refinement would be to characterize the motions of H1/head of GC1 compared to the body of the corresponding particles. We think that this is beyond the scope of this study, therefore we did not perform these experiments.

There were some instances of curious grammar/word choice that might be reconsidered by the authors, for example the use of the word vibratile/vibratility (I had to look it up!).

We have not found more appropriate scientific expressions, and thus have reformulated our sentences and replaced these terms by vibration and vibrating, which are more commonly used words.

On line 120 on page 3, “A smaller peak on” should read “A smaller peak of”. The referenced biorx manuscript is Rai, et al (not Jai et al). Page 6, line 174, “adenosin” should be “adenosine”.

We corrected these three typos in the manuscript.

Reviewer 3 Report

The maturation and assembly of the eukaryotic ribosome is the complex process that starts in the nucleolus, and followed by transport of pre-ribosomal subunits to the cytoplasm where the final maturation and formation of functional ribosome occurs. The ribosome biogenesis is a target for some drugs that have been proposed recently for cancer chemotherapy. Thus, besides the fundamental interest, the study of that process might be essential for design new anticancer medicines.
In the manuscript by Shayan et al., the authors study the poorly characterized final maturation steps of the eukaryotic small ribosomal subunit. To this end, they purified pre-40S particles from S. cerevisiae using a tagged Tsr1, a protein that is associated with immature small ribosomal subunit at stages of early and intermediate cytoplasmic particles, and performed their proteomics and cryo-EM analysis.
The main finding of this work is that during maturation in the cytoplasm, the pre-40S particle paths the stage when its beak and platform domains enter a "vibrating" state. The authors assume that the observed structural flexibility can promote the final maturation of 18S rRNA (Nob1-catalyzed cleavage of 20S pre-rRNA) and the incorporation of ribosomal proteins Rps31, Rps10 and Rps26 into the mature 40S ribosome.
In my opinion, this work provides an essential piece of structural information about a very complex process of maturation of the eukaryotic ribosome. My only criticism is related to the role of the vibrations described in this work. I understand and accept the arguments of the authors about the possible importance of flexibility in the beak and platform domains for the final maturation stage. Indeed, based on theoretical consideration, the conformational mobility of these regions may contribute to the incorporation of missing ribosomal proteins into the pre-40S particles. However, to my mind, it is hard to verify this contribution directly. The vibration can be just a side consequence of dissociation of RBFs, Ltv1 and Enp1, and it is not (maybe) necessary for the following final maturation step. Nevertheless, I understand that the authors of this work did not aim to proof or disproof the impact of the observed vibration on the formation of functional ribosomes (they are describing the fact of "vibration").
Thus, in spite of my little criticism, I am sure that this paper will be interesting for readers of your journal, and I think it should be published in Molecules.

Author Response

(Reviewer's comments are in black, authors' reply are in blue, manuscript modifications are in red).

The maturation and assembly of the eukaryotic ribosome is the complex process that starts in the nucleolus, and followed by transport of pre-ribosomal subunits to the cytoplasm where the final maturation and formation of functional ribosome occurs. The ribosome biogenesis is a target for some drugs that have been proposed recently for cancer chemotherapy. Thus, besides the fundamental interest, the study of that process might be essential for design new anticancer medicines.
In the manuscript by Shayan et al., the authors study the poorly characterized final maturation steps of the eukaryotic small ribosomal subunit. To this end, they purified pre-40S particles from S. cerevisiae using a tagged Tsr1, a protein that is associated with immature small ribosomal subunit at stages of early and intermediate cytoplasmic particles, and performed their proteomics and cryo-EM analysis.
The main finding of this work is that during maturation in the cytoplasm, the pre-40S particle paths the stage when its beak and platform domains enter a "vibrating" state. The authors assume that the observed structural flexibility can promote the final maturation of 18S rRNA (Nob1-catalyzed cleavage of 20S pre-rRNA) and the incorporation of ribosomal proteins Rps31, Rps10 and Rps26 into the mature 40S ribosome.
In my opinion, this work provides an essential piece of structural information about a very complex process of maturation of the eukaryotic ribosome. My only criticism is related to the role of the vibrations described in this work. I understand and accept the arguments of the authors about the possible importance of flexibility in the beak and platform domains for the final maturation stage. Indeed, based on theoretical consideration, the conformational mobility of these regions may contribute to the incorporation of missing ribosomal proteins into the pre-40S particles. However, to my mind, it is hard to verify this contribution directly. The vibration can be just a side consequence of dissociation of RBFs, Ltv1 and Enp1, and it is not (maybe) necessary for the following final maturation step. Nevertheless, I understand that the authors of this work did not aim to proof or disproof the impact of the observed vibration on the formation of functional ribosomes (they are describing the fact of "vibration").
Thus, in spite of my little criticism, I am sure that this paper will be interesting for readers of your journal, and I think it should be published in Molecules.

We are very thankful for Reviewer 3’s very positive comments. We fully agree with Reviewer 3 that the vibration of the beak and platform that we think we observe could only be side consequences of dissociation of RBFs/association of RPS and that it might not be strictly required for subsequent maturation events. In order to take into account the comments from all reviewers, we have significantly modified the discussion of the manuscript. We tried to better justify our observations with existing literature, while somehow simplify it too (see revised version of the manuscript for the full modifications). In the first section of the discussion, we added the following lines: “Our bottom-up proteomic data revealed that Tsr1-FPZ pre-40S particles were lacking Rps10 and Rps31, which are located on the beak of the mature 40S subunit, as well as Rps26 (on its platform). Furthermore, like in other published cytoplasmic early pre-40S cryo-EM maps, the 18S three-way junction formed by helices h34, h35 and h38 on the base of the head is in an immature conformation. We hypothesize that wriggling of the head region might be a reflect of rRNA maturation, as well as stable incorporation of “missing” RPs of the head (Rps3, Asc1, Rps10 and Rps31) of the small ribosomal subunit.

In wild type conditions, the final maturation steps of the pre-40S particle are thought to be the cleavage of the 18S rRNA 3'-end by Nob1 and the release of Dim2 and Nob1 from the platform. Dim2 release would allow the stable incorporation of Rps26 under the newly matured 18S rRNA 3’-end [13]. Here again, we speculate that platform vibration might be either a cause or a consequence to these last maturation events. The mechanisms that trigger platform wriggling remain to be established.”

We hope that these revisions, which aimed at reducing fortuitous overinterpretation of our data will fully satisfy Reviewer 3.

Round 2

Reviewer 2 Report

The authors have revised their manuscript to indicate the possibility that their results could be an artifact, which is a more open representation of their results. Given these adjustments, the manuscript is acceptable in its current form (but this reviewer still finds the use of the word "wriggling" a bit odd).